# Vicinal Risk Minimization for
# Few-Shot Cross-lingual Transfer in Abusive Language Detection

**Gretel Liz De la Peña Sarracén**
Universitat Politècnica de València
gredela@posgrado.upv.es

**Paolo Rosso**
Universitat Politècnica de València
prosso@dsic.upv.es

**Robert Litschko**[*]
MaiNLP, LMU Munich
rlitschk@cis.lmu.de

**Goran Glavaš**
CAIDAS, University of Würzburg
goran.glavas@uni-wuerzburg.de

**Simone Paolo Ponzetto**
DWS Group, University of Mannheim
ponzetto@uni-mannheim.de

## Abstract

Cross-lingual transfer learning from high-resource to medium and low-resource languages has shown encouraging results. However, the scarcity of resources in target languages remains a challenge. In this work, we resort to data augmentation and continual pre-training for domain adaptation to improve cross-lingual abusive language detection. For data augmentation, we analyze two existing techniques based on vicinal risk minimization and propose MIXAG, a novel data augmentation method which interpolates pairs of instances based on the angle of their representations. Our experiments involve seven languages typologically distinct from English and three different domains. The results reveal that the data augmentation strategies can enhance few-shot cross-lingual abusive language detection. Specifically, we observe that consistently in all target languages, MIXAG improves significantly in multidomain and multilingual environments. Finally, we show through an error analysis how the domain adaptation can favour the class of abusive texts (reducing false negatives), but at the same time, declines the precision of the abusive language detection model.

## 1 Introduction

Few-shot learning (FSL) is a machine learning paradigm that allows models to generalize from a small set of examples (Wang et al., 2020, 2023). Unlike traditional methods, FSL does not require training a model from scratch. Instead, pre-trained models are extended with just a little information, which is useful when training examples are scarce or data annotation is expensive.

Transfer learning is popularly used in few-shot learning, where the prior knowledge from a source task is transferred to the few-shot task (Pan and Yang, 2010; Pan et al., 2019). Usually, training data is abundant in the source task, while training data is low in the target task. In natural language processing, few-shot cross-lingual transfer learning (Glavaš et al., 2020; Schmidt et al., 2022; Winata et al., 2022) is the type of few-shot transfer learning in which the source/target tasks are the same but the source/target languages are different. A pre-trained multilingual model is first fine-tuned in a high-resource language and then fine-tuned on a few data in a target language (Zhao et al., 2021).

Due to the limited availability of examples in the target language, naive fine-tuning can lead to overfitting and thus poor generalization performance on the few-shot task (Parnami and Lee, 2022). A strategy usually used to alleviate this problem, not just in the few-shot cross-lingual transfer but in FSL in general, is to increase the number of samples of the few-shot task from prior knowledge. This is the data-level approach (Chen et al., 2023), which can be divided into two categories: 1) transforming samples from the few existing examples (Arthaud et al., 2021; Zhou et al., 2022; Zhang et al., 2022) and 2) transforming samples from external datasets (Antoniou and Storkey, 2019; Rosenbaum et al., 2022; Pana et al., 2023).

**Contributions.** In this work, we explore abusive language detection in seven topologically diverse languages via few-shot cross-lingual transfer learning at the data-level. Although a number of studies have examined abusive language, we aim to take advantage of resources available for English in other less explored and low-resource languages. We focus on two aspects: 1) considering languages

---

[*] Work done while at University of Mannheim

that are typologically distinct from English and 2) with little effort. Previous works focus on languages that are similar to English, such as European languages (Stappen et al., 2020; Nozza, 2021; Rodríguez et al., 2021; Firmino et al., 2021; Zia et al., 2022; Castillo-López et al., 2023). In contrast, we analyze languages that are more different from English. 'Little effort' refers to a consistent strategy across all languages, without requiring external resources or ad hoc processing for each particular language. The main contributions of this paper can be summarized as follows:

- *Dataset extension:* We rely on a multidomain and multilingual dataset for abusive language detection (Glavaš et al., 2020). This dataset contains texts in 5 languages which have been obtained by translating original English texts. To facilitate a more comprehensive evaluation, we extend the dataset by manually translating it into Spanish.

- *Few-shot cross-lingual transfer learning improvement at data-level:* We rely on Vicinal Risk Minimization (VRM) (Chapelle et al., 2000) to generate synthetic samples in the vicinity of the examples to increase the amount of information to fine-tune the model in the target language. In this work we use three VRM-based techniques: 1) SSMBA (Ng et al., 2020), which uses two functions to move randomly through a variety of data, 2) MIXUP (Zhang et al., 2018), which linearly combines pairs of examples to obtain new samples and 3) MIXAG, our variant of MIXUP, which controls the angle between an example and the synthetic data generated in its neighbourhood.

- *Unsupervised language adaptation:* We also simulate a fully unsupervised setup, removing the label information from the target languages. In that setup, we examine a strategy to address the lack of information that zero-shot transfer (no example to fine-tune the model) faces. The general idea is to make a domain adaption for abusive terms via masked language modeling (MLM) in the target language before the zero-shot transfer.

We aim to answer the following research questions:

**RQ1:** What is the role of VRM-based techniques in few-shot cross-lingual abusive language detection?

**RQ2:** What is the impact of different languages on few-shot cross-lingual abusive language detection?

**RQ3:** How does VRM-based techniques fare against domain specialization for cross-lingual transfer of abusive language detection models?

## 2 Background and Related Work

In this section, we discuss the main issue of few-shot learning and how data-based approaches can alleviate it. We take the definitions from Wang et al. (2020), where more details can be found. Then, we provide a brief overview of abusive language and align our work with recent studies focused on few-shot cross-lingual transfer approaches.

**Few-Shot Learning.** Few-shot learning deals with a small training set $D_{train} = \{(x_i, y_i)\}$ to approximate the optimal function $f^*$ that maps input $x$ to output $y$, given a joint probability distribution $p(x, y)$. Thus, a FSL algorithm is an optimization strategy that searches in a functions space $F$ to find the set of parameters that determine the best $f' \in F$. The performance is measured by a loss function $l(f(x), y)$ which defines the expected risk with respect to $p(x, y)$. However, $p(x, y)$ is unknown, hence the empirical risk is used instead (Fernandes de Mello et al., 2018). This is the average of sample losses over $D_{train}$ and can be reduced with a larger number of examples. One major challenge for FSL is then the small size of $D_{train}$, which can lead to the empirical risk not being a good approximation of the expected risk. To alleviate this problem, an approach that exploits prior knowledge can be used (Wang et al., 2023). Data-level approach involves methods that augment $D_{train}$ with prior knowledge (Feng et al., 2021; Bayer et al., 2022; Dai et al., 2023).

**Vicinal Risk Minimization** formalizes the data augmentation as an extension of $D_{train}$ by drawing samples from a neighbourhood of the existing samples (Chapelle et al., 2000). The distribution $p(x, y)$ is approximated by a vicinity distribution $D_v = \{(\hat{x}_i, \hat{y}_i)\}_{i=1}^{N_v}$, whose instances are a function of the instances of $D_{train}$. Vicinal risk ($R_v$) is then calculated on $D_v$ as Equation 1.

$$R_v = \frac{1}{N_v} \sum_{i=1}^{N_v} l(f(\hat{x}_i), \hat{y}_i) \qquad (1)$$

In this work, we study three VRM-based techniques that use different strategies to generate the vicinity distribution (see §4).

**Abusive Language.** Typically, abusive language refers to a wide range of concepts (Balayn et al., 2021; Poletto et al., 2021), including hate speech (Yin and Zubiaga, 2021; Alkomah and Ma, 2022; Jain and Sharma, 2022), profanity (Soykan et al.,

2022), aggressive language (Muti et al., 2022; Kanclerz et al., 2021), offensive language (Pradhan et al., 2020; Kogilavani et al., 2021), cyberbullying (Rosa et al., 2019) and misogyny (Shushkevich and Cardiff, 2019). Pamungkas et al. (2023) overview recent research across domains and languages. They identify that English is still the most widely studied language, but abusive language datasets have been extended to other languages, including Italian, Spanish and German (Corazza et al., 2020; Mamani-Condori and Ochoa-Luna, 2021; Risch et al., 2021). In addition, we have found studies for other languages such as Arabic (Khairy et al., 2021), Danish (Sigurbergsson and Derczynski, 2020), Dutch (Caselli et al., 2021), Hindi (Das et al., 2022), Polish (Ptaszynski et al., 2019) and Portuguese (Leite et al., 2020). Regardless, some works like (Stappen et al., 2020) state that there is a need to extend the resources for diverse and low-resource languages. To cover this problem, Glavaš et al. (2020) propose a multidomain and multilingual evaluation dataset. They show that language-adaptive additional pre-training of general-purpose multilingual models can improve the performance in transfer experiments. These are promising results, and although there are works like (Pamungkas et al., 2023) that cite this dataset, we have not found works that exploit it. In this work, we extend the study of the original work (Glavaš et al., 2020) to assess strategies for enhancing the performance of abusive language detection in low-resource languages.

**Cross-Lingual Abusive Language Detection.** In recent years, cross-lingual abusive language detection has gained increasing attention in zero-shot (Eronen et al., 2022) and few-shot (Mozafari et al., 2022) transfer. Pamungkas and Patti (2019) propose a hybrid approach with deep learning and a multilingual lexicon for cross-lingual abusive content detection. Ranasinghe and Zampieri (2020) use English data for cross-lingual contextual word embeddings and transfer learning to make predictions in languages with fewer resources. More recently, Mozafari et al. (2022) propose an approach based on meta-learning for few-shot hate speech and offensive language detection in low-resource languages. They show that meta-learning models can quickly generalize and adapt to new languages with only a few labelled data points to identify hateful or offensive content. Their meta-learning models are based on optimization-level and metric-level.

These are two approaches to improve the problem of poor data availability in few-shot learning. In contrast, we focus on the data-level approach. Unlike other works that are also based on increasing data (Shi et al., 2022), we explore VRM-based strategies for abusive language detection.

## 3 Dataset and Experimental Setup

XHate-999 (Glavaš et al., 2020) is an available dataset intended to explore several variants of abusive language detection. This dataset includes three different domains: Fox News (GAO), Twitter/Facebook (TRAC), and Wikipedia (WUL). In our work, we define ALL as the set of instances resulting from the union of all three domains. Each domain comprises different amounts of annotated data (abusive/non-abusive) in English for training, validation, and testing (see Appendix A). English test instances are translated into five target languages: Albanian (SQ), Croatian (HR), German (DE), Russian (RU), and Turkish (TR).

We extended this dataset with texts in Spanish. To generate the texts, we rely on machine translation and post-editing, following the monitored translation-based approach described in the dataset paper. Thus, slight modifications were made in the Spanish translation to reflect and maintain the level of abuse in the original English instances.

**Models.** We rely on mBERT (Devlin et al., 2019) base cased with $L = 12$ transformer layers, hidden state size of $H = 768$, and $A = 12$ self-attention heads (see Appendix A for more details). First, we retrain the model with the XHate-999 training and validation sets, to obtain the model (*model_base*) that we use in all our experiments. We search the following hyper-parameter grid: training epochs in the set $\{2, 3, 4\}$ and learning rate in $\{10^{-4}, 10^{-5}, 10^{-6}\}$. We train and evaluate in batches of 2 texts, with a maximal length of 512 tokens, and optimize the models with Adam (Kingma and Ba, 2015). We set the random seeds to 7 to facilitate the reproducibility of experiments.

**Fine-tuning and Evaluation Details.** For each language, we draw 90% of instances from the test set to evaluate *model_base*. In few-shot cross-lingual transfer experiments, we use the remaining 10% of instances to fine-tune *model_base* before the evaluation. i.e. we use 10 instances to fine-tune *model_base* in GAO (and 89 to evaluate), while the respective numbers are 30 (270)

for TRAC, 60 (540) for WUL, and 100 (899) for ALL (GAO+TRAC+WUL). Notice that for each language, the test set used by Glavaš et al. (2020) is different from the one we use. However, we do not observe a significant difference between the use of the full test set and the use of the subset we rely on (see Appendix C to examine the results).

**Statistical Analysis.** In our experiments, we used McNemar's test as (Dietterich, 1998) recommends. This is a paired non-parametric statistical hypothesis test where the rejection of the null hypothesis suggests that there is evidence to say that the models disagree in different ways. We set the significance level to 0.05 and use $\alpha_{altered}$, obtained with the Bonferroni correction (Napierala, 2012).

## 4 Few-Shot Cross-lingual Transfer

We first examine the ability of three VRM-based techniques in few-shot cross-lingual transfer learning for abusive language detection to address **RQ1**.

### 4.1 SSMBA

Ng et al. (2020) propose SSMBA, a data augmentation method for generating synthetic examples with a pair of corruption and reconstruction functions to move randomly on a data manifold. In the corruption function, we use two strategies: 1) masking a word in each text in a random way (default) or 2) masking the salient abusive words in each text. To identify abusive words, we use HurtLex (Bassignana et al., 2018), a multilingual lexicon with harmful words. For texts that do not contain words in the lexicon, we follow strategy 1. In the reconstruction functions, we use mBERT.

### 4.2 MIXUP

(Zhang et al., 2018; Sun et al., 2020) is a VRM-based technique that constructs a synthetic example $(\hat{x}_i, \hat{y}_i)$ (in the vicinity distribution) from the linear combination of two pairs $(x_i, y_i)$ and $(x_j, y_j)$, drawn at random from the training set $D_{train}$ as Equation 2, with $\lambda \sim \beta(\alpha, \alpha)$ where $\alpha$ is a hyper-parameter[1].

$$\begin{aligned} \hat{x}_i &= \lambda x_i + (1 - \lambda)x_j \\ \hat{y}_i &= \lambda y_i + (1 - \lambda)y_j \end{aligned} \quad (2)$$

We rely on a **multilingual GPT model** (Shliazhko et al., 2022) (see Appendix A) **for the linear**

combination of the texts representations (Equation 3): we obtain the embedding $E_w$ of each word of a text $x_i$ and concatenate them to generate the vector representation $E(x_i)$. Then, we combine two texts $x_i$ and $x_j$ as the linear combination of their representations $E(x_i)$ and $E(x_j)$. Note that $E_w$ is a single step of an auto-regressive model. The obtained vector is split into vectors of the same size as the original word embeddings $E_w$. Finally, we decode those vectors to obtain a sequence T of words, that we use as the new syntectic text $\hat{x}_i$. The linear combination of the labels $y \in \{0, 1\}$, when $y_i$ and $y_j$ are different depends on the value of $\lambda$. We assign 1 to $\hat{y}_i$ when the combination is greater than or equal to 0.5. Otherwise, we assign 0.

$$\hat{x}_i = T(\lambda E(x_i) + (1 - \lambda)E(x_j)) \quad (3)$$

**Procedure.** This VRM-based technique is an iterative process. In each iteration, the few-shot set $D_{train}$ is divided into pairs of samples to combine. Thus, the number of instances generated in each iteration is equal to $\frac{N}{2}$, where $N$ is the number of samples in $D_{train}$. We make sure not to take the same pairs of examples in different iterations.

### 4.3 MIXAG

Motivated by the idea of MIXUP, **we propose the variant MIXAG: mix vectors with a focus on the AnGle between them**. We hypothesize that the distance between an example and the new synthetic examples may be relevant to generate an effective vicinity. As this aspect cannot be easily controlled in the original MIXUP, we propose a particular case which interpolates pairs of instances based on the angle of their representation.

The idea is to define a linear combination (Equation 4) with the parameter $\lambda$ as a function of the angle $\alpha$ between the original vectors $x_i$ and $x_j$, as well as the angle $\theta$ between the new vector $\hat{x}$ and one of the original vectors (Figure 1).

$$\hat{x} = \lambda x_i + x_j \quad (4)$$

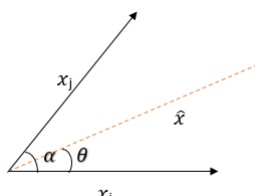

Figure 1: MIXAG description.

---

[1]We tried some values different from 1 for $\alpha$ and MIXUP was not sensitive to variation, so we set it to 0.2.

Using the Law of Sines we express $\lambda$ as a function (Equation 5) of the cosine of $\alpha$, which can be obtained with Equation 6, and the cosine of $\theta$, which is the parameter of MIXAG. $|| \cdot ||$ denotes the norm of a vector. We refer readers to Appendix B for more details.

$$\lambda = \frac{||x_j||(cos(\theta)\sqrt{1-cos(\alpha)^2}-cos(\alpha)\sqrt{1-cos(\theta)^2})}{||x_i||\sqrt{1-cos(\theta)^2}} \quad (5)$$

$$cos(\alpha) = \frac{x_i x_j}{||x_i||||x_j||} \quad (6)$$

For MIXAG, we define the combination of texts by Equation 7, following the same representation and processing of texts as in MIXUP. The difference is basically in the parameter $\lambda$.

$$\hat{x}_i = T(\lambda E(x_i) + E(x_j)) \quad (7)$$

In this work, we set $\theta = \frac{\alpha}{2}$, thus the parameter of MIXAG is defined by Equation 8. We suggest extending this study to analyze how the parameter $cos(\theta)$ can influence the results.

$$cos(\theta) = \sqrt{\frac{1+cos(\alpha)}{2}} \quad (8)$$

**Procedure.** This VRM-based technique is also an iterative process. In this case, we randomly select a sample $x_i$ from $D_{train}$ and create the pairs with $x_i$ and each of the rest of the samples of $D_{train}$. Therefore, the number of instances generated in each iteration is $N-1$, where $N$ is the number of samples in $D_{train}$.

### 4.4 Multilingual MIXUP/MIXAG

By default, in MIXUP and MIXAG we use the few-shot set $D_{train}$ of each language to generate new instances for that particular language. Alternatively, we use the union of the $D_{train}$ of all languages. For each pair of original texts $x_i$ and $x_j$, we make sure that $x_i$ is from the language in the analysis, while $x_j$ is a text from any language.

### 4.5 Multidomain MIXUP/MIXAG

We rely on training data for GAO, TRAC and WUL, as well as ALL (WUL+TRAC+GAO) in all monolingual and multilingual experiments. In short, we analyze performance when training and testing 1) only on a particular domain (for example, when testing on GAO we train only on GAO training data) and 2) on all available data from all three data sets (multidomain setup).

### 4.6 Results and Analysis

A summary of cross-lingual transfer results for the variants - few-shot and few-shot with SSMBA, MIXUP and MIXAG - is provided in Figure 2 (we refer readers to Appendix C for all the results).

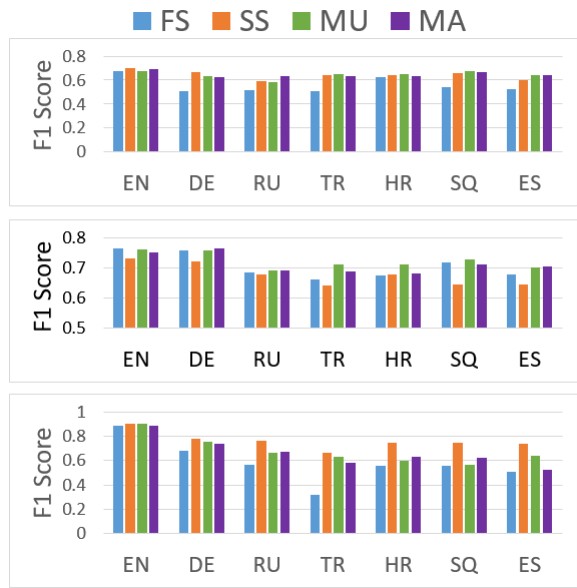

Figure 2: Performance with mBERT of few-shot (FS) cross-lingual transfer and the variants: SSMBA (SS), MIXUP (MU) and MIXAG (MA). **Upper Figure:** GAO domain, **Middle Figure:** TRAC domain and **Lower Figure:** WUL domian

As expected, we observed that VRM-based techniques improve the performance of few-shot cross-lingual transfer in most cases. There is no clear difference between the VRM-based techniques, but we can see interesting results that vary depending on the domain. In the GAO domain, all three techniques seem to have similar results across languages. In TRAC, MIXUP seems to be slightly better than MIXAG in most languages. However, the critical result in this domain is that SSMBA fails to improve the few-shot cross-lingual transfer. In contrast, SSMBA seems to be the best technique in WUL. We believe that these results are due to the nature of the texts in each domain. TRAC contains texts from Twitter and Facebook. We speculate that the reconstruction function of SSMBA affects the quality of the vicinity generated for each text by introducing terms that differ from common terms in this domain. On the other hand, WUL contains text from Wikipedia, which supports our assumption.

**Multidomain.** Table 1 shows the results for all the variants of the VRM-based techniques. We illustrate and analyze the results for the combination

| ALL | EN | DE | RU | TR | HR | SQ | ES |
|---|---|---|---|---|---|---|---|
| ZS | 0.8085 | 0.7156 | 0.6308 | 0.3627 | 0.6214 | 0.6127 | 0.6008 |
| FS | 0.8112 | 0.7141 | 0.6329 | 0.4063 | 0.6316 | 0.6238 | 0.6130 |
| SS | 0.8077 | 0.7253 | 0.7071 | 0.6568 | 0.6965 | 0.6990 | 0.6838 |
| SS-HL | 0.8097 | 0.7273 | 0.6987 | 0.6689 | 0.6725 | 0.6909 | 0.6973 |
| MU | 0.8102 | 0.7404 | 0.7013 | 0.6740 | 0.7116 | 0.7001 | 0.6878 |
| MMU | **0.8284** | 0.7500 | 0.7312 | 0.7113 | 0.7371 | 0.7128 | 0.7250 |
| MU-SS | 0.8176 | 0.7531 | 0.7233 | 0.6839 | 0.7186 | 0.6881 | 0.7087 |
| MA | 0.8083 | 0.7245 | 0.6757 | 0.5616 | 0.6710 | 0.6788 | 0.6508 |
| MMA | _0.8237_ | **0.7585** | **0.7392** | **0.7224** | **0.7523** | **0.7344** | **0.7476** |
| MA-SS | 0.8096 | 0.7229 | 0.7193 | 0.6369 | 0.6759 | 0.6734 | 0.6713 |

Table 1: Zero-shot (ZS) and few-shot (FS) cross-lingual transfer performance with mBERT on the union of all domains. We also show 8 variants for FS: 1) SSMBA (SS) and 2) SSMBA with HurtLex (SS-HL), 3) MIXUP (MU), 4) multilingual MIXUP (MMU), 5) MIXUP with SSMBA (MU-SS), 6) MIXAG (MA), 7) multilingual MIXAG (MMA), 8) MIXAG with SSMBA (MA-SS). The results ($\alpha_{altered} = .005$) are reported in terms of F1 and significantly better results are underlined for each language and domain. Numbers in bold indicate the best results.

of all domains. The results by domain are detailed in Appendix C.

All languages except German seem to benefit from few-shot cross-lingual transfer w.r.t. zero-shot cross-lingual transfer. Likewise, the few-shot cross-lingual transfer is improved with VRM-based techniques as in the results by domain.

SSMBA improves few-shot cross-lingual transfer in all languages except English. In this heterogeneous domain, we do not observe the problem that SSMBA has in TRAC. On the other hand, the use of HurtLex does not seem to be a relevant strategy, since the results are similar to those obtained with the default strategy (random selection). This is an encouraging result, which suggests that we can use SSMBA to improve few-shot cross-lingual transfer learning without relying on external resources.

MIXUP seems to be better than SSMBA and MIXAG for most languages. However, multilingual MIXAG is significantly the best strategy. This is a good indicator of the benefits of our variant for multidomain and multilingual environments. Note that the multilingual strategies outperform the rest of the variants and that particularly, multilingual MIXAG consistently performs better than multilingual MIXUP. This suggests that our hypothesis about the implication of controlling the angle between the original texts and the new synthetic texts seems to be relevant in multilingual data.

Finally, we combine MIXUP/MIXAG with SSMBA: First, we augment the data with SSMBA and then augment the new vicinity with MIXUP/MIXAG. The results are also shown in Table 1. This strategy offers some improvement over MIXUP/MIXAG in most cases.

**Correlation Analysis.** Thus far, we have observed that the behaviour of the strategies seems quite similar across languages. For instance, the few-shot cross-lingual transfer is outperformed with the VRM-based techniques. This motivates us to investigate **RQ2**, i.e. we examine if there is a high correlation between the performance of few-shot cross-lingual transfer (and its variants with VRM-based techniques) and the linguistic proximity scores of each language to English.

We analyze the correlation between the performance of the strategies that we use for cross-lingual transfer learning and the distance between each language and English. We rely on the tool LANG2VEC[2] which proves language vectors that encode linguistic features from the URIEL database (Littell et al., 2017). We obtain the vector representation of the languages with 4 features: 1) SYN: encodes syntactic properties, 2) FAM: encodes memberships in language families, 3) INV: denotes the presence of natural classes of sounds and 4) PHO: encodes phonological properties.

Then, with the vectors from each linguistic feature, we calculate the cosine similarity between each language and English. Finally, we calculate the Pearson correlation coefficients (Sedgwick, 2012) between the cosine similarity and the performance of each cross-lingual strategy across languages and domains.

Table 2 shows the correlation coefficients for the significant linguistic features with a significance

[2]https://github.com/antonisa/lang2vec

|     | SYN   | FAM   | INV   | PHO   |
| --- | ----- | ----- | ----- | ----- |
| FS  | 0.664 | 0.661 | 0.607 | 0.516 |
| SS  | 0.527 | 0.627 | 0.608 | 0.486 |
| MU  | 0.405 | 0.628 | 0.633 | 0.463 |
| MA  | 0.571 | 0.721 | 0.686 | 0.529 |

Table 2: Pearson correlation coefficients between linguistic proximity scores (features SYN, FAM, INV, PHO) and few-shot (FS), few-shot with SSMBA (SS), few-shot with MIXUP (MU) and few-shot with MIXAG (MA) cross-lingual transfer performance with mBERT across all languages and domains.

level of 0.05 (Appendix C shows the correlation coefficients for all metrics and the similarity scores between each language and English). Coefficients whose magnitude is between 0.5 and 0.7 indicate a moderate correlation, while coefficients between 0.3 and 0.5 indicate a low correlation.

We only observe a moderate correlation between the performance of each strategy and the distance between the target languages and English. We consider these results encouraging because they suggest that the strategies are possibly consistent across languages.

## 4.7 Ablation Studies

MIXAG is a data augmentation method that randomly combines inputs and accordingly combines one-hot-label encodings. This is a variant of MIXUP where the new data is obtained by defining the angle between the inputs and the new instance.

In our strategy, we randomly select pairs of inputs and set the angle between the new instance and one of the inputs as $\theta = \frac{\alpha}{2}$, where $\alpha$ is the angle between the original inputs. However, there are other strategies that could be used. For example, selecting data pairs whose latent representations are close neighbors, as well as defining other values for $\theta$. To compare MIXAG with these alternative possibilities, we run a set of ablation study experiments using not only mBERT, but also the XLM-R model (Conneau et al., 2020). We focus on multilingual and multimodal MIXAG (MMA in ALL) as it is the best data augmentation method that we observed in the first experiments.

On the one hand, we compare the combination of random pairs of inputs with the combination of nearest neighbors (NN). On the other hand, we set the angle $\theta = \frac{\alpha}{3}$ to evaluate the impact of varying this parameter on the performance of the method. Finally, we use an alternative model for the text rep-

resentation. Specifically, we used the multilingual generative model mT0 (Muennighoff et al., 2023), instead of mGPT.

From the results of the ablation study in Table 3, we have the following observations. First, there are no significant differences with $\alpha = .05$ between the variants studied, although experiments with XLM-R seem to have shown some improvement. Secondly, we note that the variation of the angle between the inputs and the generated instances does not seem to represent a relevant factor.

All five variants obtain very similar results with mBERT. The variation of the factors that we analyze does not seem to influence the performance of the method. However, with XLM-R we observe some interesting findings. Spanish and Russian are the only languages where MMA method is not surpassed by the other variants. In the rest of the languages, we observe the opposite behaviour, where text representation with the alternative model mT0 seems to be the best strategy. Notice that in Albanian the use of mT0 for text representation together with the strategy of selecting the nearest neighbor for interpolation seems to be the best variant.

## 5 Unsupervised Language Adaptation

In this section, we investigate the scenarios in which there is no information about the target language for the few-shot cross-lingual transfer. In § 4 we used a small amount of supervised data $D_{train}$ in the target language to fine-tune the pre-trained model. This allowed us to adapt the model to the abusive language of each particular language. In contrast, now we assume that the labels of $D_{train}$ are not available. This is a simulated experiment where we only have an unlabelled set of texts and the set $D_{test}$ in which we want to detect abusive language. Previous works have examined this scenario by adjusting a model with unlabelled external data. In this work, **we use only a few unlabelled instances from $D_{train}$.**

Basically, this strategy is a zero-shot cross-lingual transfer learning in which the model is adapted to the abusive terms of the target language. As mBERT is pre-trained on general-purpose and multilingual corpora, it is familiar with the target languages. However, it has not been adjusted to the particular case of abusive language. We follow then a two-step methodology: 1) continual pre-taining for domain adaptation via masked language modeling (MLM) to make it familiar to the partic-

| model | variant | EN | DE | RU | TR | HR | SQ | ES |
|-------|---------|------|------|------|------|------|------|------|
| mBERT | MMA | 0.8237 | 0.7585 | 0.7392 | 0.7224 | 0.7523 | 0.7344 | 0.7476 |
| | MMA-NN | 0.8233 | 0.7585 | 0.7201 | 0.6473 | 0.7523 | 0.7273 | 0.7466 |
| | MMA-ANG | 0.8233 | 0.7475 | 0.7169 | 0.6774 | 0.7415 | 0.7273 | 0.7466 |
| | MMA-MT0 | 0.8238 | 0.7585 | 0.7392 | 0.7224 | 0.7523 | 0.7344 | 0.7476 |
| | MMA-MT0-NN | 0.8254 | 0.7687 | 0.7314 | 0.6696 | 0.7477 | 0.7314 | 0.7528 |
| XLM-R | MMA | 0.8236 | 0.7927 | 0.7561 | 0.7258 | 0.7180 | 0.7780 | 0.7670 |
| | MMA-NN | 0.8251 | 0.7942 | 0.7328 | 0.7267 | 0.7180 | 0.7797 | 0.7650 |
| | MMA-ANG | 0.8251 | 0.7940 | 0.7521 | 0.7267 | 0.7216 | 0.7797 | 0.7650 |
| | MMA-MT0 | 0.8245 | 0.7952 | 0.7503 | 0.7281 | 0.7243 | 0.7798 | 0.7658 |
| | MMA-MT0-NN | 0.8245 | 0.7940 | 0.7503 | 0.7297 | 0.7180 | 0.7803 | 0.7658 |

Table 3: Results of the ablation studies for 4 variants of multilingual MIXAG (MMA): 1) interpolation only between nearest neighbors (MMA-NN), 2) set $\theta = \frac{\alpha}{3}$ (MMA-ANG), 3) text representation with mT0 (MMA-MT0) and 4) text representation with mT0 and interpolation between nearest neighbors(MMA-MT0-NN). The results ($\alpha_{altered} = .005$) are reported in terms of F1 for each of the model mBERT and XLM-R.

ular abusive terms, and then 2) employ zero-shot learning to detect abusive language.

## 5.1 Results and Analysis

Table 4 illustrates the results obtained with the methodology across domains and languages. In most cases, the strategy of prior adaptation to the abusive terms seems to outperform zero-shot cross-lingual transfer learning. English is the only language in which the MLM adaptation worsens the results in all domains. Moreover, TRAC also shows no improvement, similar to the behaviour observed with SSMBA in few-shot cross-lingual transfer.

These results allow us to answer **RQ3**: although domain adaptation can improve zero-shot cross-lingual transfer, VRM-based techniques seem to be more robust in few-shot cross-lingual transfer.

**Error Analysis.** In order to deepen the analysis of what happens in the model with the zero-shot cross-lingual transfer adaptation, we also analyze two metrics: Recall and Precision. Recall refers to the true positive rate and is the number of true positives divided by the total number of positive texts. Precision refers to the positive predictive value and is the number of true positives divided by the total number of positive predictions. In this work, positive refers to the class of abusive texts.

Results across domains and languages are in Appendix C. In all cases we observe an increase in Recall, indicating that adapting the model could improve the proportion of the class of abusive texts that is correctly classified. At first glance, it seems to be a good result, since it is desirable to reduce the number of false negatives in abusive language detection. However, we observe that precision is reduced, suggesting that this strategy favours the positive class: while false negatives are reduced, false positives are increased.

Critical cases are negative texts that can be incorrectly detected as abusive. In order to study this phenomenon, we examine the percentage of texts that are non-abusive and are well-classified with zero-shot transfer learning and misclassified with the MLM adaptation. We investigate two statistics across languages and domains: 1) the percentage of non-abusive texts that are well- classified with zero-shot transfer and misclassified with the MLM adaptation and 2) the percentage of abusive texts that are misclassified with zero-shot transfer and well-classified with the MLM adaptation.

Table 5 illustrates the statistics across domains and languages. Consistent with the previous results we observe a detriment in the class of non-abusive texts. The number of negative texts well-classified with zero-shot transfer learning and misclassified with the MLM adaptation is large (reaching 100% in a case). However, that amount is surpassed in most cases by the gain in the class of abusive texts. We observe that the number of positive texts that are misclassified with zero-shot transfer learning and well-classified with adaptation via MLM is high (reaching 100% in four cases).

## 6 Conclusions and Future Work

In this work, **we studied three techniques to improve few-shot cross-lingual transfer learning in abusive language detection**. These techniques are concentrated on data-level approach to deal with the problem of data scarcity that can lead to a high estimation error in few-shot learning. Specifically, **we focused on vicinal risk minimization techniques** to increase the data in the vicinity of the

| GAO | EN | DE | RU | TR | HR | SQ | ES |
|---|---|---|---|---|---|---|---|
| ZS | **0.6747** | 0.5067 | 0.5205 | 0.5116 | 0.6234 | 0.5405 | 0.5263 |
| ZS_MLM | 0.6050 | **0.6364** | **0.6261** | **0.6341** | **0.6290** | **0.6016** | **0.6154** |
| **TRAC** | | | | | | | |
| ZS | **0.7642** | **0.7582** | **0.6815** | 0.6777 | **0.6892** | **0.7235** | **0.7000** |
| ZS_MLM | 0.6821 | 0.6480 | 0.6718 | **0.6785** | 0.6785 | 0.6995 | 0.6118 |
| **WUL** | | | | | | | |
| ZS | **0.8800** | 0.6698 | 0.5561 | 0.2945 | 0.5469 | 0.5556 | 0.4960 |
| ZS_MLM | 0.6093 | **0.6765** | **0.6708** | **0.6765** | **0.6732** | **0.6675** | **0.6765** |
| **ALL** | | | | | | | |
| ZS | **0.8085** | **0.7156** | 0.6308 | 0.3627 | 0.6214 | 0.6127 | 0.6008 |
| ZS_MLM | 0.6662 | 0.6711 | **0.6637** | **0.6716** | **0.6716** | **0.6721** | **0.6419** |

Table 4: Zero-shot (ZS) and adapted zero-shot (ZS_MLM) cross-lingual transfer performance with mBERT on domains (GAO, TRAC, WUL) and the union of all domains (ALL). Results are reported in terms of F1 and numbers in bold indicate those that are significantly better for each language and domain ($\alpha = .05$).

| GAO | EN | | DE | | RU | | TR | | HR | | SQ | | ES | |
|---|---|---|---|---|---|---|---|---|---|---|---|---|---|---|
| | %P | %N | %P | %N | %P | %N | %P | %N | %P | %N | %P | %N | %P | %N |
| **GAO** | 83.3 | 60 | 76.1 | 61.7 | 85.7 | 77.7 | 100 | 80.7 | 81.2 | 86.4 | 85.0 | 88.8 | 100 | 97.1 |
| **TRAC** | 81.2 | 74.6 | 82.3 | 83.8 | 91.8 | 88.8 | 88.1 | 92.9 | 87.5 | 80.7 | 100 | 90.5 | 69.2 | 89.3 |
| **WUL** | 85.7 | 83.8 | 62.2 | 78.0 | 56.8 | 63.8 | 74.5 | 89.3 | 88.3 | 96.1 | 99.3 | 98.8 | 96.7 | 98.8 |
| **ALL** | 87.2 | 83.9 | 76.6 | 84.1 | 100 | 100 | 100 | 88.4 | 88.9 | 94.5 | 99.5 | 99.4 | 83.9 | 93.3 |

Table 5: Percentage of non-abusive texts that are well-classified with zero-shot transfer and misclassified with the MLM adaptation (%N), and percentage of abusive texts that are misclassified with zero-shot transfer and well-classified with the MLM adaptation (%P).

few-shot samples. First, we explored two existing techniques: 1) SSMBA, which is based on a pair of functions to corrupt and reconstruct texts, and 2) MIXUP, which generates new samples from a linear combination of original instances pairs. Then, **we proposed MIXAG, a variant of MIXUP, to parameterize the combination of instances with the angle between them**. Our experiments were based on the multidomain and multilingual dataset XHATE-999, which allowed us to explore low-resource languages as target languages and English as the base language. This dataset contains six different languages, and we extended it to Spanish, following the same methodology that was used to generate the texts of the other languages. The results showed the effectiveness of VRM-based techniques to improve few-shot cross-lingual transfer learning in most domains and languages. Particularly, we observed that multilingual MIXAG outperforms the other strategies in the heterogeneous set (multidomain) for all target languages. At the same time, we observed that structural language similarity does not seem to be highly correlated with cross-lingual transfer success in none of the strategies. These results are encouraging for abusive language detection in low-resource settings, as the strategies that we have examined appear to be consistent across languages.

Finally, we evaluated a scenario where it is not possible to perform a few-shot cross-lingual transfer due to the lack of supervised information. We used a strategy based on masked language modeling and saw a degradation in the class of non-abusive texts, but a gain in the class of abusive texts, reducing false negatives.

In future work, we aim to further examine our proposed VRM-based technique for data augmentation. MIXAG uses as a parameter the angle between the new instance and one of the original instances being combined. In our experiments, we fixed the angle as half the angle between the original instances, but we consider that the flexibility of varying that parameter must be exploited.

# 7 Limitations and Ethical Concerns

Our experiments relied on a dataset that only contains English texts in the training and development sets. Only the test set is multilingual. Therefore, we were forced to partition the test set in order to perform the few-shot cross-lingual transfer and domain adaptation experiments. We compared the

results obtained in zero-shot cross-lingual transfer with the original test set and with the subset used in our experiments. We did not observe statistical differences. However, this may be a limitation in comparing our results with the original results reported in the dataset paper. Moreover, we observed a limitation in the strategy of domain adaptation. As we discussed in the error analysis, although the class of abusive texts is favoured with this strategy, we observed a detriment in the negative class.

This work aims to improve abusive language detection in low-resource languages. While this can be useful for many languages, there are certain ethical implications. Therefore, we strongly recommend not using the proposed strategies as the sole basis for decision-making in abusive language detection. Regarding the issue of privacy, all the data we use in our experiments, both the original dataset and the new texts in Spanish that we generated, are publicly available. It should be noted that the scope of this work is strictly limited to the evaluation of models that are also publicly available, and it is not used to promote abusive language with the information obtained.

## Acknowledgements

FairTransNLP research project (PID2021-124361OB-C31) funded by MCIN/AEI/10.13039/501100011033 and by ERDF, EU A way of making Europe.

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

## A  Reproducibility

Table 6 provides features and links to the pre-trained models that we use, and Table 7 illustrates details of the dataset.

| Model: | mBERT |
|---|---|
| Vocab size: | 120k |
| #Params: | 177M |
| Link: | https://huggingface.co/bert-base-multilingual-cased |
| Use in this work: | SSMBA |
| | Experiments (zero-shot and few-shot cross-lingual transfer) |
| Model: | mGPT |
| Vocab size: | 100k |
| #Params: | 1417M |
| Link: | https://huggingface.co/ai-forever/mGPT |
| Use in this work: | MIXUP & MIXAG |
| Model: | XLM-R |
| Vocab size: | 250k |
| #Params: | 270M |
| Link: | https://huggingface.co/xlm-roberta-base |
| Use in this work: | Ablation studies |
| Model: | mT0 |
| Vocab size: | 250k |
| #Params: | 550M |
| Link: | https://huggingface.co/bigscience/mt0-base |
| Use in this work: | Ablation studies |

Table 6: Features of the models used in this work. We built our models directly on top of the HuggingFace Transformers library.

| Domain | Train (EN) | Validation (EN) | Test (LANG) |
|---|---|---|---|
| **GAO** | 919 | 218 | 99 |
| **TRAC** | 10,341 | 2,593 | 300 |
| **WUL** | 71,754 | 24,130 | 600 |
| **ALL** | 83,014 | 26,941 | 999 |

Table 7: Number of texts in each set of the XHate-999 dataset. LANG stands for each language in {EN, SQ, HR, DE, RU, TR, ES}.

## B  MIXAG Details

MIXAG is a particular case of MIXUP where the parameter $\lambda$ of the linear combination (Equation 9) is determined by the angle $\alpha$ between the original vectors $x_i$ and $x_j$, as well as the angle $\theta$ between the new vector $\hat{x}$ and one of the original vectors. We take $x_i$ without loss of generality (see Figure 3). We rely on the cosine of $\alpha$, calculated as Equation 10, where $|| \cdot ||$ denotes the norm of a vector. Notice that we only parameterize one of the original vectors, since $\alpha$ and $\theta$ are sufficient to determine $\hat{x}$.

$$\hat{x} = \lambda x_i + x_j \tag{9}$$

$$cos(\alpha) = \frac{x_i x_j}{||x_i|| ||x_j||} \tag{10}$$

The objective is to express the parameter $\lambda$ as a function of $\theta$, hence we take advantage of the Law of Sines (Equation 11) that allows relating

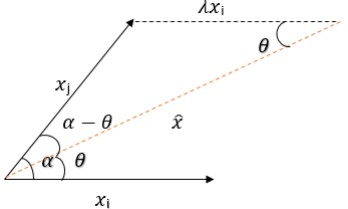

Figure 3: MIXAG explanation.

vectors and angles. Then, $\lambda$ can be expressed in function of $\theta$ as Equation 12. Finally, using the known identities in Equations 13, we can define $\lambda$ from the cosine of $\alpha$, which can be obtained with Equation 10, and the cosine of $\theta$, which is the parameter of MIXAG (Equation 14).

$$\frac{\lambda ||x_i||}{sin(\alpha-\theta)} = \frac{||x_j||}{sin(\theta)} \tag{11}$$

$$\lambda = \frac{||x_j|| sin(\alpha-\theta)}{||x_i|| sin(\theta)} \tag{12}$$

$$sin(\alpha-\theta) = sin(\alpha)cos(\theta) - cos(\alpha)sin(\theta)$$
$$sin(\theta) = \sqrt{1-cos(\theta)^2}, \quad sin(\alpha) = \sqrt{1-cos(\alpha)^2}$$
$$sin(\alpha-\theta) = \sqrt{1-cos(\alpha)^2}cos(\theta) - cos(\alpha)\sqrt{1-cos(\theta)^2} \tag{13}$$

$$\lambda = \frac{||x_j||(cos(\theta)\sqrt{1-cos(\alpha)^2} - cos(\alpha)\sqrt{1-cos(\theta)^2})}{||x_i||\sqrt{1-cos(\theta)^2}} \tag{14}$$

## C  Results by Language and Domain

We show complete results in this section. Table 8 illustrates that there is no significant difference between using the full test set and using a subset of texts from the test set (the subset that we used in our experiment).

Table 9 illustrates the cosine similarity between each language and English for five linguistic features. We obtain these features as language vectors from LANG2VEC (Littell et al., 2017).

Table 10 shows the correlation coefficient and p-value for these linguistic features.

- **SYN:** vectors encode syntactic properties, e.g., if a subject appears before or after a verb.

- **FAM:** vectors encode memberships in language families.

- **INV:** vectors denote the presence or absence of natural classes of sounds.

- **PHO:** vectors encode phonological properties such as the consonant-vowel ratio.

| GAO | EN | DE | RU | TR | HR | SQ | ES |
|------|------|------|------|------|------|------|------|
| FZS | 0.6742 | 0.5185 | 0.5063 | 0.5217 | 0.6098 | 0.5432 | 0.5060 |
| ZS | 0.6747 | 0.5067 | 0.5205 | 0.5116 | 0.6234 | 0.5405 | 0.5263 |
| **TRAC** | | | | | | | |
| FZS | 0.7594 | 0.7527 | 0.6859 | 0.6806 | 0.6925 | 0.7177 | 0.7045 |
| ZS | 0.7642 | 0.7582 | 0.6815 | 0.6777 | 0.6892 | 0.7235 | 0.7000 |
| **WUL** | | | | | | | |
| FZS | 0.8812 | 0.6739 | 0.5581 | 0.2969 | 0.5476 | 0.5675 | 0.5049 |
| ZS | 0.8800 | 0.6698 | 0.5561 | 0.2945 | 0.5469 | 0.5556 | 0.4960 |
| **ALL** | | | | | | | |
| FZS | 0.8053 | 0.7146 | 0.6322 | 0.3565 | 0.6231 | 0.6088 | 0.6028 |
| ZS | 0.8085 | 0.7156 | 0.6308 | 0.3627 | 0.6214 | 0.6127 | 0.6008 |

Table 8: Cross-lingual transfer performance with mBERT on each domain (GAO, TRAC, WUL) and the union of all domains (ALL). FZS refers to zero-shot cross-lingual transfer with the full test set, which corresponds to the results reported in XHATE-999: Analyzing and Detecting Abusive Language Across Domains and Languages. ZS refers to zero-shot cross-lingual transfer with 90% of the test set of each language, which corresponds to the results discussed in this paper. Results are reported in terms of F1.

| | EN | DE | RU | TR | HR | SQ | ES |
|------|------|------|------|------|------|------|------|
| SYN | 1.0 | 0.9025 | 0.8118 | 0.5067 | 0.8318 | 0.7959 | 0.8216 |
| FAM | 1.0 | 0.5443 | 0.1667 | 0.0 | 0.1260 | 0.3333 | 0.0962 |
| INV | 1.0 | 0.7628 | 0.6475 | 0.6658 | 0.6967 | 0.7249 | 0.6382 |
| PHO | 1.0 | 0.8058 | 0.8581 | 0.8181 | 0.8581 | 0.8704 | 0.8581 |
| GEO | 1.0 | 0.9976 | 0.9681 | 0.9825 | 0.9950 | 0.9919 | 0.9959 |

Table 9: Cosine similarity between each language vector and English vector for LANG2VEC-based language vectors (SYN, FAM, INV) considering all domains.

- **GEO:** vectors express orthodromic distances for languages w.r.t. fixed points on the Earth's surface.

Table 11 shows the Precision and Recall results across domains and languages for the error analysis of the unsupervised language adaptation.

Table 12 shows the results for all variants across languages and domains.

|     | SYN | | FAM | | INV | | PHO | | GEO | |
| --- | --- | --- | --- | --- | --- | --- | --- | --- | --- | --- |
|     | Pearson | P-value | Pearson | P-value | Pearson | P-value | Pearson | P-value | Pearson | P-value |
| ZS | 0.672 | <.001 | 0.647 | <.001 | 0.599 | <.001 | 0.529 | .003 | 0.302 | .119 |
| FS | 0.664 | <.001 | 0.661 | <.001 | 0.607 | <.001 | 0.516 | .004 | 0.289 | .136 |
| SS | 0.527 | .004 | 0.627 | <.001 | 0.608 | <.001 | 0.486 | .008 | 0.261 | .180 |
| MU | 0.405 | .033 | 0.628 | <.001 | 0.633 | <.001 | 0.463 | .013 | 0.315 | .210 |
| MA | 0.571 | .001 | 0.721 | <.001 | 0.686 | <.001 | 0.529 | .004 | 0.245 | .209 |

Table 10: Complete table of correlations between zero-shot (ZS), few-shot (FS), few-shot with SSMBA (SS), few-shot with MIXUP (MU) and few-shot with MIXAG (MA) cross-lingual transfer performance with mBERT across all languages and domains, with linguistic proximity scores (features SYN, FAM, INV, PHO, GEO). Correlations that are not statistically significant are underlined ($\alpha$=.05).

| GAO | EN | | DE | | RU | | TR | | HR | | SQ | | ES | |
| --- | --- | --- | --- | --- | --- | --- | --- | --- | --- | --- | --- | --- | --- | --- |
|     | R | P | R | P | R | P | R | P | R | P | R | P | R | P |
| ZS | 0.70 | 0.65 | 0.48 | 0.54 | 0.48 | 0.58 | 0.55 | 0.48 | 0.60 | 0.65 | 0.50 | 0.59 | 0.50 | 0.55 |
| ZS_MLM | 0.80 | 0.47 | 0.88 | 0.50 | 0.78 | 0.50 | 0.98 | 0.47 | 0.98 | 0.46 | 0.93 | 0.45 | 0.98 | 0.44 |
| **TRAC** | | | | | | | | | | | | | | |
| ZS | 0.89 | 0.67 | 0.88 | 0.66 | 0.74 | 0.63 | 0.71 | 0.65 | 0.78 | 0.62 | 0.85 | 0.63 | 0.73 | 0.67 |
| ZS_MLM | 0.92 | 0.54 | 0.81 | 0.54 | 0.92 | 0.53 | 0.93 | 0.53 | 0.93 | 0.53 | 0.98 | 0.54 | 0.72 | 0.53 |
| **WUL** | | | | | | | | | | | | | | |
| ZS | 0.80 | 0.98 | 0.51 | 0.97 | 0.39 | 0.94 | 0.17 | 0.96 | 0.38 | 0.97 | 0.40 | 0.92 | 0.33 | 0.97 |
| ZS_MLM | 0.83 | 0.48 | 0.89 | 0.49 | 0.99 | 0.51 | 0.98 | 0.51 | 0.99 | 0.51 | 0.97 | 0.51 | 0.99 | 0.51 |
| **ALL** | | | | | | | | | | | | | | |
| ZS | 0.79 | 0.82 | 0.71 | 0.72 | 0.57 | 0.71 | 0.24 | 0.75 | 0.56 | 0.69 | 0.54 | 0.71 | 0.51 | 0.74 |
| ZS_MLM | 0.99 | 0.50 | 0.99 | 0.51 | 0.98 | 0.50 | 0.96 | 0.50 | 0.99 | 0.51 | 0.99 | 0.51 | 0.89 | 0.50 |

Table 11: Precision (P) and Recall (R) in cross-lingual transfer with mBERT on each domain (GAO, TRAC, WUL) and the union of all domains (ALL). ZS_MLM refers to adapted zero-shot cross-lingual transfer, while ZS refers to zero-shot cross-lingual transfer.

| GAO | EN (n) | DE (n) | RU (n) | TR (n) | HR (n) | SQ (n) | ES (n) |
|---|---|---|---|---|---|---|---|
| ZS | 0.6747 | 0.5067 | 0.5205 | 0.5116 | 0.6234 | 0.5405 | 0.5263 |
| FS | 0.6500 | 0.5455 | 0.5455 | 0.5556 | 0.6000 | 0.5385 | 0.5385 |
| SS | **0.7000** | 0.6667 | 0.5952 | 0.6400 | 0.6452 | 0.6598 | 0.6024 |
| SS-HL | 0.6667 | **0.6737** | 0.6136 | 0.6408 | 0.6588 | 0.6667 | 0.6279 |
| MU | 0.6742 | 0.6316 | 0.5806 | 0.6481 | 0.6531 | 0.6742 | 0.6444 |
| MMU | 0.6667 | 0.6316 | 0.5500 | 0.6355 | 0.6588 | **0.6882** | 0.6471 |
| MU-SS | 0.6591 | 0.6122 | 0.5926 | 0.6304 | 0.6444 | 0.6667 | 0.5333 |
| MA | 0.6882 | 0.6214 | 0.6304 | 0.6364 | 0.6292 | 0.6667 | 0.6408 |
| MMA | 0.6889 | 0.6526 | **0.6383** | **0.6549** | **0.6667** | 0.6517 | **0.6517** |
| MA-SS | **0.7033** | 0.6408 | 0.5977 | 0.6471 | 0.6444 | 0.6465 | 0.5278 |
| **TRAC** | | | | | | | |
| ZS | **0.7642** | 0.7582 | 0.6815 | 0.6777 | 0.6892 | 0.7235 | 0.7000 |
| FS | 0.7462 | 0.7561 | 0.6861 | 0.6620 | 0.6731 | 0.7183 | 0.6782 |
| SS | 0.7303 | 0.7208 | 0.6775 | 0.6411 | 0.6783 | 0.6452 | 0.6444 |
| SS-HL | 0.7261 | 0.7124 | 0.6776 | 0.6282 | 0.6736 | 0.6498 | 0.6544 |
| MU | 0.7613 | 0.7576 | 0.6923 | **0.7103** | 0.7103 | **0.7290** | 0.7003 |
| MMU | 0.7596 | 0.7508 | 0.6894 | 0.6906 | 0.6796 | 0.7148 | 0.7043 |
| MU-SS | 0.7403 | 0.7325 | **0.7010** | 0.6709 | **0.7162** | 0.6220 | 0.6894 |
| MA | 0.7516 | **0.7622** | 0.6901 | 0.6885 | 0.6801 | 0.7093 | **0.7047** |
| MMA | 0.7516 | **0.7622** | 0.6943 | 0.6885 | 0.6935 | 0.7066 | **0.7047** |
| MA-SS | 0.7222 | 0.7190 | 0.6554 | 0.6596 | 0.6567 | 0.6596 | 0.4909 |
| **WUL** | | | | | | | |
| ZS | 0.8800 | 0.6698 | 0.5561 | 0.2945 | 0.5469 | 0.5556 | 0.4960 |
| FS | 0.8845 | 0.6761 | 0.5657 | 0.3193 | 0.5567 | 0.5833 | 0.5040 |
| SS | 0.9035 | 0.7821 | **0.7636** | 0.6620 | **0.7443** | 0.7490 | 0.7348 |
| SS-HL | 0.9014 | 0.7696 | 0.7059 | 0.6636 | 0.7007 | 0.7043 | 0.7043 |
| MU | 0.9035 | 0.7576 | 0.6605 | 0.6333 | 0.5950 | 0.5650 | 0.6357 |
| MMU | 0.9006 | 0.7495 | 0.6574 | 0.6029 | 0.6394 | 0.5985 | 0.5802 |
| MU-SS | **0.9077** | 0.7776 | 0.7140 | **0.6800** | 0.7245 | 0.7071 | **0.7495** |
| MA | 0.8822 | 0.7406 | 0.6758 | 0.5821 | 0.6344 | 0.6230 | 0.5277 |
| MMA | 0.8911 | 0.7350 | 0.6605 | 0.5260 | 0.6682 | 0.6502 | 0.5656 |
| MA-SS | 0.9014 | **0.7911** | 0.7149 | 0.6787 | 0.7315 | **0.7505** | 0.7228 |
| **ALL** | | | | | | | |
| ZS | 0.8085 | 0.7156 | 0.6308 | 0.3627 | 0.6214 | 0.6127 | 0.6008 |
| FS | 0.8112 | 0.7141 | 0.6329 | 0.4063 | 0.6316 | 0.6238 | 0.6130 |
| SS | 0.8077 | 0.7253 | 0.7071 | 0.6568 | 0.6965 | 0.6990 | 0.6838 |
| SS-HL | 0.8097 | 0.7273 | 0.6987 | 0.6689 | 0.6725 | 0.6909 | 0.6973 |
| MU | 0.8102 | 0.7404 | 0.7013 | 0.6740 | 0.7116 | 0.7001 | 0.6878 |
| MMU | **0.8284** | 0.7500 | 0.7312 | 0.7113 | 0.7371 | 0.7128 | 0.7250 |
| MU-SS | 0.8176 | 0.7531 | 0.7233 | 0.6839 | 0.7186 | 0.6881 | 0.7087 |
| MA | 0.8083 | 0.7245 | 0.6757 | 0.5616 | 0.6710 | 0.6788 | 0.6508 |
| MMA | 0.8237 | **0.7585** | **0.7392** | **0.7224** | **0.7523** | **0.7344** | **0.7476** |
| MA-SS | 0.8096 | 0.7229 | 0.7193 | 0.6369 | 0.6759 | 0.6734 | 0.6713 |

Table 12: Zero-shot (ZS) and few-shot (FS) cross-lingual transfer performance with mBERT on each domain (GAO, TRAC, WUL) and the union of all domains (ALL). We also show 8 variants for FS: 1) SSMBA (SS) and 2) SSMBA with HurtLex (SS-HL), 3) MIXUP (MU), 4) multilingual MIXUP (MMU), 5) MIXUP with SSMBA (MU-SS), 6) MIXAG (MA), 7) multilingual MIXAG (MMA), 8) MIXAG with SSMBA (MA-SS). The results are reported in terms of F1 and significantly better results are underlined for each language and domain. Numbers in bold indicate the best results. We use $\alpha_{altered} = .005$ since we have 10 tests