# OpenReview forum: "Vicinal Risk Minimization for Few-Shot Cross-lingual Transfer in Abusive Language Detection"
_EMNLP/2023/Conference — EMNLP 2023 Main_

### Official Review · Reviewer_mJE9 · 2023-08-04

**Soundness:** 3

**Excitement:**

4: Strong: This paper deepens the understanding of some phenomenon or lowers the barriers to an existing research direction.

**Paper Topic And Main Contributions:**

This paper proposes a new technique for cross-lingual abusive language detection. It exploits large available datasets in English to perform cross-lingual transfer across seven completely different languages from English and across three domains.

**Questions For The Authors:**

1. The answer to RQ3 depends on the assumption that MLM could be used as a technique to let any language be aware of the abusive terms used. Then why doesn’t it work for English? This either illustrated this assumption is flawed or it does not properly answer RQ3.
2. It is difficult to understand whether MLM adaptation works better than zero-shot transfer. For some domain MLM works better, whereas in other domains zero-shot transfer works better. Can you clarify what are trying to entail by comparing them?

**Reasons To Accept:**

1. The paper proposes an incremental method MIXAG over two existing data augmentation methods: SSMBA and MIXUP.
2. The lambda parameter in MIXUP is selected using a beta distribution over a hyperparameter. In MIXAG, lambda is computed using the candidate data points and their angle and vicinity. This approach is simple and works well in practice.
3. Extensive correlation analysis is performed to demonstrate that the used strategies work well across all the languages.

**Reasons To Reject:**

1. Quantitative results are not convincing to prove the superiority of the proposed MIXAG approach. Firstly, the author states that there is not much difference between existing VRM-based techniques which is also reflected in Figure
2. Across domains and languages, it is not conclusive whether MIXUP performs better than MIXAG or vice-versa.
3. Authors say that Multilingual MIXAG outperforms all the baselines when union of all the domains are used. It is not clearly evaluated multilingual MIXAG outperforms other setups. Also, please clearly write what you intend to convey with Figure 2 and Table 1 respectively.

**Reproducibility:**

3: Could reproduce the results with some difficulty. The settings of parameters are underspecified or subjectively determined; the training/evaluation data are not widely available.

**Reviewer Confidence:**

4: Quite sure. I tried to check the important points carefully. It's unlikely, though conceivable, that I missed something that should affect my ratings.

---

> ### Author Rebuttal · Authors · 2023-08-29
>
> We thank the Reviewer 3 for reviewing our work. We appreciate your comments and hope the following comments may clarify your concerns:
>
> > Quantitative results are not convincing to prove the superiority of the proposed MIXAG approach. Firstly, the author states that there is not much difference between existing VRM-based techniques which is also reflected in Figure
>
> We propose MIXAG, a variant of MIXUP, whose main contribution is in the multidomain and multilingual environment. On the one hand, Table 1 illustrates the results in the multidomain environment, where we show that the multilingual MIXAG (MMA) improves significantly (underlined values) to other techniques. On the other hand, Figure 2 summarizes the results for each domain, where there are no significant differences between the VRM techniques, as noted by the reviewer. This is why we point out that there are no significant differences between the techniques.
>
> > Across domains and languages, it is not conclusive whether MIXUP performs better than MIXAG or vice-versa.
>
> RQ3 focuses on the comparison of MLM domain adaptation against VRM techniques for cross-lingual transfer of abusive language detection. We found that VRM techniques performed better than zero-shot transfer, which we did not observe for MLM domain adaptation. For example, the case of English, cited by the reviewer, was always better with zero-shot transfer than with domain adaptation. Therefore, the conclusion for RQ3 is that VRM techniques seem to be more robust than MLM domain adaptation and consistently perform better across all languages. Note, that searching for strategies to improve MLM in each particular language is beyond the scope of this paper.
>
> > Authors say that Multilingual MIXAG outperforms all the baselines when union of all the domains are used. It is not clearly evaluated multilingual MIXAG outperforms other setups. Also, please clearly write what you intend to convey with Figure 2 and Table 1 respectively.
>
> The comparison of MLM domain adaptation against zero-shot transfer in Table 3 is to show that MLM fails to improve zero-shot transfer in many cases. These results are used to analyze RQ3, as mentioned in the previous point.

---

### Official Review · Reviewer_zEh7 · 2023-08-04

**Soundness:** 3

**Excitement:**

3: Ambivalent: It has merits (e.g., it reports state-of-the-art results, the idea is nice), but there are key weaknesses (e.g., it describes incremental work), and it can significantly benefit from another round of revision. However, I won't object to accepting it if my co-reviewers champion it.

**Paper Topic And Main Contributions:**

This paper proposed to use data augmentation and continual pre-training for domain adaptation to improve cross-lingual abusive language detection. Specifically:

* The paper proposed MIXAG, a vicinal risk minimization-based data augmentation method.
* Their experiments showed that MIXAG improves significantly in multi-domain and multi-lingual environments.

**Questions For The Authors:**

1. Why VRM? There are many other data augmentation methods of similar nature.

**Reasons To Accept:**

* The paper achieved decent performance improvements.
* The paper experimented with topologically diverse languages.
* The paper is mostly clear about its contributions, methods, and results.

**Reasons To Reject:**

* Many of the decisions made in the experimental settings are well-motivated. For example, why VRM? Why GPT for the combination? These are choices that need to be made on either a theoretical or an empirical basis.
* Some of the important findings are left in the appendices, like appendix C. This paper needs to be restructured.

**Reproducibility:**

2: Would be hard pressed to reproduce the results. The contribution depends on data that are simply not available outside the author's institution or consortium; not enough details are provided.

**Reviewer Confidence:**

4: Quite sure. I tried to check the important points carefully. It's unlikely, though conceivable, that I missed something that should affect my ratings.

**Typos Grammar Style And Presentation Improvements:**

Please don't use bar graphs. We only get a rough estimate of the F1 score in the bar graph. Make a table and put in the numbers.

---

> ### Author Rebuttal · Authors · 2023-08-29
>
> We thank the Reviewer 2 for reviewing our paper. We appreciate your comments and hope the following comments may clarify your concerns:
>
> 1. The VRM principle is an empirical risk minimization variant based on vicinal functions. There is strong evidence of improvements with VRM in terms of generalization with vicinal functions. MIXUP is a popular choice of vicinal distribution that introduces globally linear behaviour between training examples and improves the generalization performance of models. Many works have shown that models trained on MIXUP are relatively robust to input perturbations. Motivated by these findings, we aim to leverage this VRM technique to address cross-lingual transfer in abusive language detection with scarce data. At the same time, we propose and evaluate MIXAG, a variant of MIXUP, which shows improvements in a multidomain and multilingual environment.
>
> 2. We plan to include a section on ablation studies in a new version of the article. Among the aspects to be considered is the representation of texts for the combination. Note that this will allow us to analyze the empirical basis of the chosen model, as the reviewer points out.
>
> 3. We agree that it is clearer for the reader if the results are presented in a table with the numerical values for each experiment. In fact, our first intention was to present the results only in tables, but the large number of experiments made the table really large, as you can see in Appendix C. As a strategy, we decided to summarize the results in a graph that can help analyze the results by domain. Note that the main contributions of MIXAG are in the multidomain and multilingual environment. Therefore, we ensure that the results of greatest interest (multidomain) are detailed in Table 1.
> Due to space limitations, we included many detailed results in Appendix C. However, note that we summarized the most important content that supports our conclusions and refer to the specific places in the appendices for more details.

---

### Official Review · Reviewer_bbqD · 2023-08-07

**Soundness:** 3

**Excitement:**

3: Ambivalent: It has merits (e.g., it reports state-of-the-art results, the idea is nice), but there are key weaknesses (e.g., it describes incremental work), and it can significantly benefit from another round of revision. However, I won't object to accepting it if my co-reviewers champion it.

**Paper Topic And Main Contributions:**

This paper focuses on cross-lingual abusive language detection and addresses the challenge of resource scarcity in target languages. The main contributions are:

Data Augmentation: The paper proposes a novel data augmentation method called MIXAG, which interpolates pairs of instances based on the angle of their representations. This technique, along with analyzing existing vicinal risk minimization-based methods, enhances cross-lingual abusive language detection.

Domain Adaptation: The researchers utilize continual pre-training for domain adaptation, aiming to improve model performance in diverse linguistic and domain contexts.

Overall, the paper's contributions lie in proposing MIXAG, leveraging data augmentation and domain adaptation to enhance cross-lingual abusive language detection. The results validate the effectiveness of the techniques in addressing resource scarcity and improving performance across diverse linguistic and domain settings.

**Reasons To Accept:**

Strengths of the paper:

Novel Data Augmentation Technique: The paper introduces a novel data augmentation method called MIXAG, which interpolates pairs of instances based on the angle of their representations. This approach provides a fresh perspective on data augmentation for cross-lingual abusive language detection.

Addressing Resource Scarcity: The paper addresses the challenge of resource scarcity in target languages, which is a common issue in cross-lingual NLP. By utilizing data augmentation and domain adaptation, the work offers practical solutions for improving model performance in low-resource settings.

**Reasons To Reject:**

While the paper has several strengths, it may also have some weaknesses that could be considered for further improvement:

Limited Comparison with Baselines: The paper does not provide a detailed comparison with existing baseline methods for data augmentation and domain adaptation in cross-lingual text classification. A thorough comparison with state-of-the-art techniques, such as  XLM-R, would strengthen the validity and significance of the proposed approaches.

Lack of Extensive Ablation Studies: The paper does not conduct extensive ablation studies to analyze the individual contributions of data augmentation and domain adaptation to the overall performance improvement.

Interpretability and Analysis of MIXAG: The paper could include more in-depth analysis and explanation of the novel data augmentation method (MIXAG). Understanding how MIXAG affects the representations and contributes to the model's performance would add clarity and insights into the approach.

**Reproducibility:**

3: Could reproduce the results with some difficulty. The settings of parameters are underspecified or subjectively determined; the training/evaluation data are not widely available.

**Reviewer Confidence:**

4: Quite sure. I tried to check the important points carefully. It's unlikely, though conceivable, that I missed something that should affect my ratings.

---

> ### Author Rebuttal · Authors · 2023-08-29
>
> We thank the Reviewer 1 for the reviews --  we appreciate the comments and hope the following replies may soothe some of the concerns raised:
>
> > Limited Comparison with Baselines: The paper does not provide a detailed comparison with existing baseline methods for data augmentation and domain adaptation in cross-lingual text classification. A thorough comparison with state-of-the-art techniques, such as XLM-R, would strengthen the validity and significance of the proposed approaches.
>
> Vicinal risk minimization formalizes data augmentation, where a neighbourhood is defined around each example in the training data. Then, additional synthetic examples can be drawn from the vicinity distribution of training examples to increase the support of the training distribution. Most data augmentation techniques assume that the examples in the neighbourhood belong to the same class.
>
> MIXUP is a data augmentation strategy based on linear interpolations of feature vectors and their associated targets, which can be from different classes. This strategy has been shown to increase the robustness of neural networks and improve generalization. We start from these fundamentals and propose MIXAG, whose strengths have been observed in multidomain and multilingual environments.
>
> With MIXAG as a data augmentation strategy, we would argue that the most meaningful baselines are other established data augmentation strategies. We thus use MIXUP, a highly popular and cited augmentation strategy for dense representation spaces, as a primary baseline. We also use SSMBA, another VRM technique that focuses on text processing. We believe that these are the most appropriate baselines.
>
> Note that all augmentation strategies -- our MIXAG, as well as the baselines -- operate on top of the same multilingual LM (mBERT). While we agree that repeating the experiments and findings on top of another multilingual LM such as XLM-R would make sense and strengthen our work, this does not mean that we do not compare against strong augmentation baselines (i.e., MIXUP).
>
> Post-hoc, indeed, we considered XLM-R in our experiments. We observed that these results did not lead to significant differences in our conclusions (same trends as for mBERT). Therefore, given the limitation of space in the paper, we decided not to add these results. We will gladly add the results of both our MIXAG and the baselines on top of XLM-r, given the extra space for the camera ready version of the paper.
>
> > Lack of Extensive Ablation Studies: The paper does not conduct extensive ablation studies to analyze the individual contributions of data augmentation and domain adaptation to the overall performance improvement.
>
> MIXAG is a data augmentation method that randomly combines inputs and accordingly combines one-hot-label encodings.
> With the possibility of increasing the size of the paper, we intend to add a discussion of different design options. However, we would like to emphasize that MIXAG is not a complex comosite approach for which ablations over individual components are obvious.
>
> Some of the options one could consider (again, these are not ablations in the traditional sense of ablating the components of a multi-component approach):
>   - Interpolate only between nearest neighbors instead of random selection, or only between inputs of the same class.
>   - Use different text representation strategies (e.g., from instruction fine-tuned generative multilingual models, e.g., mT0).
>   - When generating a new instance from two original inputs, we set the angle to half the angle between these original inputs. In the new section, we will add the results obtained when varying this parameter.
>
> > Interpretability and Analysis of MIXAG: The paper could include more in-depth analysis and explanation of the novel data augmentation method (MIXAG). Understanding how MIXAG affects the representations and contributes to the model's performance would add clarity and insights into the approach.
>
> In this regard, we plan to include an illustration showing the representation of original instances and the synthetic instances created with MIXAG (i.e., to plot the topology of the representation space and see where the synthetic instances position themselves w.r.t. original ones). This will allow us to show how this strategy manages to match the statistics of the training set, especially in a multilingual setting. Note that in a data augmentation method, there is a desire for the new data to match the distribution of the training set. Using augmented data that deviates too much from the distribution of the training set can increase the training error in the original training set, which negatively affects generalization.

---

### Meta-Review · Area_Chair_M3hX · 2023-09-19

**Recommendation:** 5

**Metareview:**

This paper focuses on cross-lingual few-shot abusive language detection and addresses the scarcity of target language data by data augmentation and further pre-training for domain adaptation. Based on the existing vicinal risk minimization methods (SSMBA, MIXUP), the authors propose a new data augmentation method, MIXAG, that uses the angle between text representations to interpolate them. Furthermore, they test continue pretraining on a few unlabelled examples and compare it with the zero-shot baseline.

As mentioned by the reviewers, the authors introduce a novel data augmentation method MIXAG, and show its effectiveness with decent performance improvements in few-shot cross-lingual transfer especially in the multilingual scenario (Multilingual MIXUP). Although results from the multidomain setting (Figure 2) do not show superior performance, the authors contextualized these results in their rebuttal and pointed out the impact of their approach in multidomain and multilingual environments.

I believe the additional content that the authors mentioned during the discussion period, such as XLM-R results and analysis on the representation of original and synthetic instances, will further improve the paper.

---

### Decision · Program_Chairs · 2023-10-07

**Decision:**

Accept-Main

**Comment:**

This paper focuses on cross-lingual few-shot abusive language detection and addresses the scarcity of target language data by data augmentation and further pre-training for domain adaptation. Based on the existing vicinal risk minimization methods (SSMBA, MIXUP), the authors propose a new data augmentation method, MIXAG, that uses the angle between text representations to interpolate them. Furthermore, they test continue pretraining on a few unlabelled examples and compare it with the zero-shot baseline.

As mentioned by the reviewers, the authors introduce a novel data augmentation method MIXAG, and show its effectiveness with decent performance improvements in few-shot cross-lingual transfer especially in the multilingual scenario (Multilingual MIXUP). Although results from the multidomain setting (Figure 2) do not show superior performance, the authors contextualized these results in their rebuttal and pointed out the impact of their approach in multidomain and multilingual environments.

I believe the additional content that the authors mentioned during the discussion period, such as XLM-R results and analysis on the representation of original and synthetic instances, will further improve the paper.